# Time-Efficient Identification Procedure for Neurological Complications of Rescue Patients in an Emergency Scenario Using Hardware-Accelerated Artificial Intelligence Models

Abu Shad Ahammed *, Aniebiet Micheal Ezekiel and Roman Obermaisser *

Chair of Embedded System, Department of Electrical Engineering and Computer Science, University of Siegen, 57076 Siegen, Germany; micheal.ezekiel@uni-siegen.de
* Correspondence: abu.ahammed@uni-siegen.de (A.S.A.); roman.obermaisser@uni-siegen.de (R.O.)

**Abstract:** During an emergency rescue operation, rescuers have to deal with many different health complications like cardiovascular, respiratory, neurological, psychiatric, etc. The identification process of the common health complications in rescue events is not very difficult or time-consuming because the health vital symptoms or primary observations are enough to identify, but it is quite difficult with some complications related to neurology e.g., schizophrenia, epilepsy with non-motor seizures, or retrograde amnesia because they cannot be identified with the trend of health vital data. The symptoms have a wide spectrum and are often non-distinguishable from other types of complications. Further, waiting for results from medical tests like MRI and ECG is time-consuming and not suitable for emergency cases where a quick treatment path is an obvious necessity after the diagnosis. In this paper, we present a novel solution for overcoming these challenges by employing artificial intelligence (AI) models in the diagnostic procedure of neurological complications in rescue situations. The novelty lies in the procedure of generating input features from raw rescue data used in AI models, as the data are not like traditional clinical data collected from hospital repositories. Rather, the data were gathered directly from more than 200,000 rescue cases and required natural language processing techniques to extract meaningful information. A step-by-step analysis of developing multiple AI models that can facilitate the fast identification of neurological complications, in general, is presented in this paper. Advanced data analytics are used to analyze the complete record of 273,183 rescue events in a duration of almost 10 years, including rescuers' analysis of the complications and their diagnostic methods. To develop the detection model, seven different machine learning algorithms-Support Vector Machine (SVM), Random Forest (RF), K-nearest neighbor (KNN), Extreme Gradient Boosting (XGB), Logistic Regression (LR), Naive Bayes (NB) and Artificial Neural Network (ANN) were used. Observing the model's performance, we conclude that the neural network and extreme gradient boosting show the best performance in terms of selected evaluation criteria. To utilize this result in practical scenarios, the paper also depicts the possibility of embedding such machine learning models in hardware like FPGA. The goal is to achieve fast detection results, which is a primary requirement in any rescue mission. An inference time analysis of the selected ML models and VTA AI accelerator of Apache-TVM machine learning compiler used for the FPGA is also presented in this research.

**Keywords:** neurology; machine learning; TVM; VTA; artificial intelligence; support vector machine; random forest; logistic regression; naive bayes; artificial neural network; K-nearest neighbour; prediction; rescue patient



## 1. Introduction

Our brain is a complex organ of the human body, also known as the command center for the human nervous system playing a role incontrolling memory, emotion, body movements, and even the basic human functions like breathing or vision. It remains till

today a mystery to us how our brain consistently generates control commands ranging from sensing and locomotion to emotion, decision-making, learning, and memory [1]. Any dysfunction or disorder of these control processes due to adverse effects on the brain or nervous system leads to neurological complications. Henceme of the common neurological diseases like epilepsy, cerebral palsy, Alzheimer, apoplexy, and brain stroke affect more than a billion people worldwide while approximately seven million people every year die as a result of these diseases [2]. Another study from 2016 [3] suggests that neurological complications are the leading causes of disability among humans and the second leading cause of death every year. A person having neurological disorders can be affected in many forms including disability in walking, speaking, learning, and moving from one place to another [4]. As these disorders are quite lethal and can affect one adversely for the rest of their life of, a necessity of early detection is quite important, especially when the patient is in an emergency rescue situation [5]. Currently, emergency rescuers have no predefined procedure for diagnosing neurological complications in rescue patients. For detecting if a patient had stroke-related issues, the F.A.S.T. (Facial drooping, Arm weakness, Speech difficulties and Time) assessment test is performed. However, this test is not effective for other neurological complications like epilepsy or intercostal neuralgia, as the symptoms can go away after some time or are not at all noticeable. It may also happen that the symptoms visible in a patient can belong to other similar complications like psychiatric diseases. Based on our discussion with the rescue station in Siegen–Wittgenstein, the initial impression from the rescuers on patients plays a vital role in the diagnosis of neurological complications. In emergency scenarios, where a single wrong decision can result in fatal consequences, the current diagnostic procedure is not effective. A good solution to this problem can be the use of artificial intelligence in rescue situations, which not only can improve the diagnostic procedures for neurological diseases, but also be time efficient.

Digitalization of healthcare has opened up the potential for generating and collecting huge amounts of biomedical data of a patient using sensors and electronic devices. State-of-the-art data analysis tools and computing devices make it feasible to deepen our knowledge of individual health data, but the availability of such data comes with different challenges for the data researchers, as often it is found that the data are high dimensional, unstructured, redundant, or partly quite ambiguous to be able to draw insight into patient's health. Similar challenges were confronted while working with our rescue dataset, as often the data were incomplete, redundant and often provide no meaningful information in raw form. This is not unexpected because during a rescue mission, it is quite challenging to collect data and make a diagnosis in a limited time frame. Hence, some unwanted artifacts and contaminated information are present in the dataset which needs to be filtered for extracting meaningful information.

For neurological patients, to understand their health prognosis with common symptoms and vital parameters, it is particularly important to go through rescuers' initial health analysis in past cases and look for specific keywords pertaining to this complication. Natural language processing tools were quite resourceful in that aspect after we selected the text attributes in the dataset and extracted new features relevant to neurological diseases.

Machine learning (ML) is a subset of artificial intelligence which has already proved its effectiveness in the diagnosis and prediction of diverse incurable diseases at an earlier stage [6]. The purpose of using machine learning in the health sector is to increase the efficiency of medical treatment by detecting if a patient has some specific disease based on his/her health pattern. Developing such a model can be very resourceful in the case of rescue missions because a delay in detecting health complications during emergencies causes obstacles to initiating necessary treatment paths, resulting in a high mortality rate. Besides, the traditional diagnosis methods applied in rescue situation has several drawbacks like [7]:

1.  Lengthy diagnostic process which involves many observations and a wide range of questions
2.  Requirement of high expertise from rescuers

3. Decision-making based on the available data requires a lot of time and can come out wrong

4. Due to inadequate data, no correct diagnosis can be done, and a patient is just referred to the hospital

Here, we must not forget that the experience of physicians to diagnose such critical complications is undeniable and often irreplaceable. The use of ML algorithms is only to make the diagnostic process smoother, more reliable and more precise.

The objective of this research is two fold. Firstly, we develop detection models that are time efficient and provide good accuracy to overcome the challenges caused by traditional diagnostic methods as explained above. Different machine learning algorithms are chosen for the detection model to test their performance in detecting neurological complications of rescue patients. Further, their time to infer complication probability from an unknown health pattern is considered. Prior to model development, analysis of neurological rescue cases is done with different data analysis techniques. The analysis facilitates extracting new features to create a training dataset, which is an essential part of machine learning algorithms. This paper will provide an overview of the detection models developed with the dataset from the rescue station, Siegen–Wittgenstein of their rescue missions from 2012 till 2021. Seven different algorithms including neural network are used to develop the detection models and later a performance comparison in terms of accuracy and time efficiency is provided.

There are a handful of examples using machine learning in the diagnosis of health complications like cardiovascular, neurological or cancer detection using X-ray, Magnetic resonance imaging (MRI), computed tomography angiogram (CTA), radiology imaging tests or pathology reports. The novelty of this research is, here the dataset considered for the detection model is generated only from health vitals collected in rescue operations and primary observation from rescuers. No medical test data or pathology were used because it is not practical in emergency situations to wait for hours to process the test results. Even in many rescue cases, those medical tests cannot be conducted due to the limitation of test equipment or time-contingency. The power of machine learning to derive predictive models from substantial amounts of rescue data with minimal or, even in some cases, without providing prior knowledge of the data can play a vital role in this scenario.

The model developed is accelerated in a Xilinx FPGA to find out the model's suitability in real-world situations. FPGA shows tremendous potential in the case of accelerating neural network because of its feature of reconfiguration ability of hardware architecture down to bit-level, fast computational capability and doing massive parallel operations. Hence, in rescue situations where a quick inference is highly expected after passing the health data of emergency patients, FPGA is an innovative solution. However, the purpose of this paper is not to provide the technical aspects of machine learning model acceleration in FPGA or its extensive implementation. The paper will elaborate briefly on the challenges of working with rescue cases belonging to neurological complications and how such challenges were resolved with innovative data analysis to develop an optimized detection model.

The rest of the manuscript is organized as follows. Section 2 provides a meta-analysis of the existing research works, methods and techniques by different authors. Section 3 will discuss the methodology applied to perform deep data analysis of rescue records and extract key features for developing training dataset. Section 4 will discuss briefly on the machine learning algorithms used to develop the detection model. An overview of the acceleration of machine learning model in FPGA using a compiler like tensor virtual machine (TVM) and AI-accelerator versatile tensor accelerator(VTA) is reviewed in Section 5. Section 6 discusses the performance of the detection models in terms of accuracy and time efficiency after providing an overview of selected evaluation criteria. Potential limitations and challenges faced during our research are also discussed in this section. Finally, Section 7 will provide a conclusion, mentioning the existing challenges and a look at the possibilities in the future.

## 2. Related Researches

In this section, we will review and summarize published literature on the application of machine learning to detect neurological complications in general. Another topic of discussion will be how TVM and VTA have performed previously when used as tools to deploy machine learning algorithms in FPGA. Several search engines and repositories, e.g., Google Scholar, IEEE Explorer, MDPI, and PubMed are used to identify existing and relevant articles. After a comprehensive review of the literature retrieved initially, we found that despite the current success of machine learning in detection models for many scientific areas including health complications, the eminence of machine learning in neurological complications for rescue patients, is yet to be well explored. The data type we are using, i.e., rescue data are unheard of in this area of research as per our exploration.

The meta-analysis of neurological diseases diagnosis with deep learning presented by Gautam R. et al., 2020 [8] has explored 136 research articles to confer the discipline, frameworks, and methodologies used by different deep learning techniques to diagnose different human neurological disorders like stroke, Alzheimer, Parkinson's, epilepsy, autism, migraine, cerebral palsy, and multiple sclerosis. The goal of that research was to analyze and examine the performance and publication trend of different deep learning techniques employed in the investigation of these diseases. The research study showed that classification accuracy in the diagnosis of particular neurological and neuropsychiatric disorders lies in the range between 90 and 100%. Diagnosis of Alzheimer's disorder has the highest rate of accuracy, i.e., 100% using a convolution neural network. The researchers concluded that the scope of such diagnosis should be extended with deep learning to achieve more effective performance.

A. A.-A.Valliani et al., 2019 [9] provided a systematic review of employing deep learning in the clinical neuroscience field with the existing challenges and a look to the future. The research highlighted the broad spectrum of medical image-based data, their classification, segmentation, and their effectiveness in risk prognostication. The performance of machine learning models using non-image data like electroencephalography is also reviewed stating their high sensitivity and specificity for epileptic seizure detection. The authors concluded that for improved diagnosis or early detection of acute neurological events, various domains of deep learning can play a significant role, especially when the patient is suffering from AD, ASD, or ADHD. It was also emphasized that to understand the true potential of deep learning, two major challenges need to be overcome. Firstly, while the technical challenges surrounding the generalizability and interpretability of machine learning models exist, more difficult challenges like data privacy and accessibility are important. The second challenge is the data quality, which is an identical problem to our research, as the data collected through rescue procedures are often very raw and require a lot of filtering before using in a model. Other research like that [10] from Afshin S. et. al. provided a comprehensive overview of works focused on specific neurological disorders, in this case, automated epileptic seizure detection using deep learning techniques and neuroimaging modalities. A performance-based review of different machine learning algorithms for detecting early falls of elderly persons was provided in [11] by Nahiduzzaman M. The author used recurrent neural network, random forest and support vector machine to develop the detection model which provided an accuracy of 96–98%, but no significant research was found that works with rescue data to detect neurological complications in general which is the prime goal of this research. One of the authors of this paper Ahammed, A.S., in a different research paper [12] used the rescue data to develop several machine learning models for the identification of respiratory complications in emergency patients. The accuracy of the finally selected model was 91% which motivated us to research further on neurological complications for rescue patients.

Not so many significant studies were also found on the machine learning compiler Apache-TVM and AI accelerator VTA. The key contributors of Apache-TVM have provided a thorough review of it in [13], while summarizing the challenges faced during deep learning model acceleration and optimization at the computation graph level and tensor

operator level. The challenges are: High-level dataflow rewriting, Memory reuse across threads, Tensorized compute intrinsic, and Latency hiding. The authors explained in the paper how using two optimization layers: a computation graph optimization layer and a tensor optimization layer with new schedule primitives TVM resolves the challenges. In another paper [14] published in 2019, we got an overview of the VTA hardware–software stack built into the TVM. An end-to-end performance evaluation over multiple CPU, GPU, and FPGA-equipped edge systems was analyzed. The developers stated that for comparable systems, VTA provides a significant performance edge over conventional CPU and GPU-based inference.

## 3. Materials and Methods

### 3.1. Study Population

The data used in this research were collected through a research project—'KIRETT' where the aim is to provide early medical treatment for rescue patients after detecting their complications. Rescue station, Siegen–Wittgenstein recorded this data from January 2012 till August 2021 in a city of Germany called Siegen at different rescue scenarios. These 10 years of historical database of past rescue attempts consist of 273,183 unique cases identified with numerous complication types and more than 400 unique attributes. Both quantitative and categorical data are present in the database. Due to the privacy issue, the data handed over was anonymous. The rescue cases represent the history of rescue operations for different patients suffering from complications like cardiovascular, respiratory, neurological, psychiatric, abdominal, digestive, etc. Each of these cases was recorded with information-like parameters of the rescue, geographical information of the operation, the patient's medical history, health diagnosis, vital signs, first impression of the complication, medications administered, and treatment executed. From the health diagnosis attributes, we have categorized the complication type of a patient. In Figure 1, an overview of several patients with different complication types is presented based on diagnostic information.

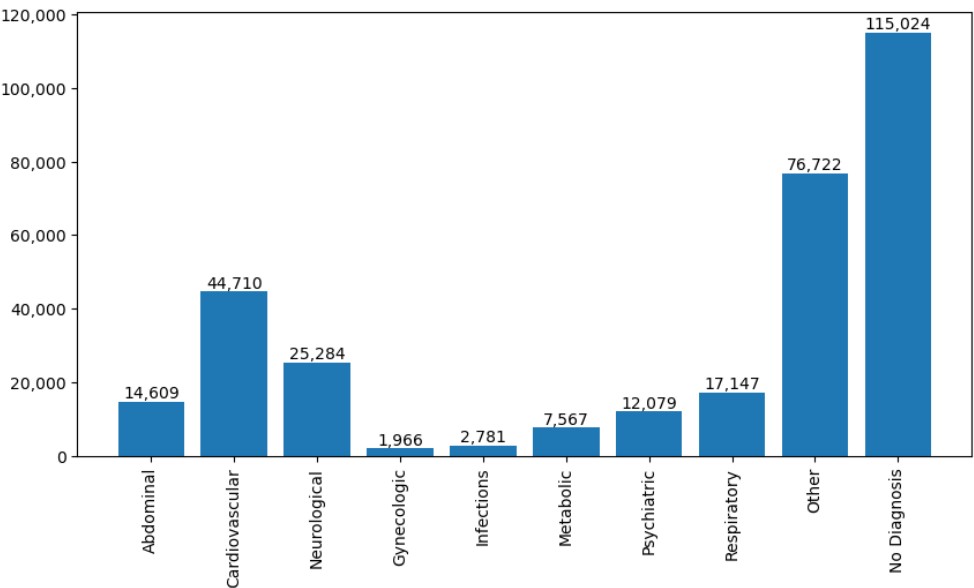

**Figure 1.** Number of rescue patients with different types of health complications as per the 10 years record of the Rescue station, Siegen–Wittgenstein. Many rescue events were found without proper diagnostic information, and were thereby uncategorized.

We have used python based data analysis tool to parse through the record of diagnostic data and collected the statistics as shown in Figure 1. The number of rescue patients found with neurological complications is 25,284. The five most common neurological diseases

we observed from the database are: trauma, concussion, apoplex, brain stroke and seizure attack. Henceme of the records were diagnosed as a general neurological disorder and some have a close connection with psychoneurotic disease. We have considered all those cases in our initial dataset to diversify the information spectrum of neurological diseases.

*3.2. Proposed Methodology*

In the data part, the methodology we used for this research consists of three major steps: data management, machine learning model development, and performance evaluation. Figure 2 shows the steps in the proposed workflow which involves the pre-processing of data by sorting and filtering, text parsing, feature engineering to create a training dataset, development of ML model and performance evaluation. This work is implemented in Python 3.

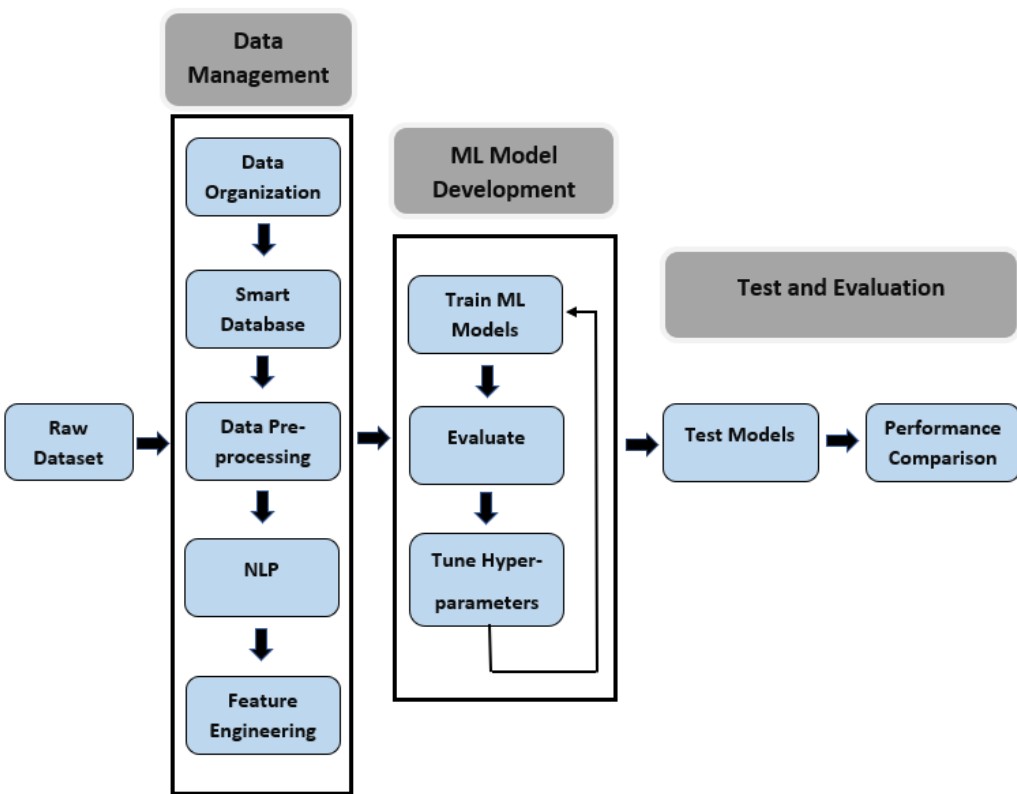

**Figure 2.** Proposed Methodology on Data Analysis and ML Model Development.

At the initial phase, we had to develop a deep data analysis procedure because the primary data were not usable and full of outliers in raw form. The first and most crucial step was to organize the raw data and manage it in a smart database for faster inquiry. After managing the dataset, we emphasized pre-processing the data and doing feature engineering for collecting relevant attributes of neurological disease and creating the training dataset. Natural language processing techniques were also applied to the dataset for extracting new features that come out from rescuers' assessments. As mentioned in [15], the performance of a machine learning model is upper bounded by the data quality. Afterward, we chose different machine learning algorithms with a grid of hyperparameters to evaluate which parameters are the most suitable in terms of obtaining better detection performance. For testing purposes, we used Python function train_test_split for generating test dataset and later checked how the models perform on those unseen test data. To visualize and summarize the performance of each model, different performance metrics were calculated based on the confusion matrix. The results indicate the suitability of each model if deployed to diagnose real-world samples.

### 3.3. Data Organization

Data initially received from the rescue station was collected through 80 different files in HTML format. Much repetitive and ambiguous information was present in the dataset without any significance for the research. Hence, after an initial data sorting, we removed some attributes like the address of the patients, type of rescue wagon, timestamp information, after-rescue hospital details, or type of treatment given to a patient. A data dictionary software was developed using Python to decipher rescue-related words and record them in an organized way for future reference. The dictionary also explains all the acronyms and data types within the database. We also created a Python-based algorithm to merge all the records provided by the rescue station in a single file, where redundant data columns were removed. We considered the rescue case IDs as master keys and combined unique information of duplicate cases in their respective data cells. The raw database contained a lot of null values and sometimes included information that has no significant importance for the research.

### 3.4. Database Management

An open-source object-relational database management system (ORDBMS) like PostgreSQL was used for our rescue database to manage the database smartly. ORDBMS is used because of its ability to query complex applications preserving the original data structure and handling large and complex applications, which is a requirement in our case [16]. This database management system can serve as a subsidiary to practically all SQL operations. It also has a reliable architecture with dependability, data integrity and precision [17]. Another reason for choosing PostgreSQL is, it exhibits a higher quality of performance when the environment requires a precise and structured data model [18]. After organizing the raw data, we migrated it to a PostgreSQL database using the client software pgAdmin 4. PgAdmin is considered the most popular and feature-enriched open-source administration development tool with a supported graphics user interface (GUI). The GUI can be used to interact with the PostgreSQL database sessions, both locally and on remote servers as well. The rescue station data were initially grouped into 18 different tables based on the relevance of the information. To make efficient data analysis, the tables were segregated from the original data and put inside query-enabled tables using the client software.

### 3.5. Data Pre-Processing

Data pre-processing is known as a data mining technique, where the goal is to manipulate and drop data after detecting anomalies in order to ensure or enhance the data quality. The common 4 steps for data pre-processing are: data fusion, data cleaning, data structuring, and data summarizing which we have performed on our dataset [19]. The data from the rescue station was recorded in an emergency environment where the collection of information is quite difficult and sometimes gets distorted. Hence, it was not surprising to find misspelled text information, and abnormal values of health vitals like blood glucose, respiratory rate, etc. It initiated the necessity to perform data pre-processing techniques to make the data usable for machine learning applications.

The data pre-processing was performed in both horizontal and vertical directions. In horizontal pre-processing, we searched for rescue cases that belong to neurological complications. There were attributes with diagnostic information of the patients written by rescuers in the dataset. Henceme of this information was collected directly from the patients or from their families that describe various health complaints pertaining to neurology. While gathering such information, it became critical in many cases as the diagnostic was referring to multiple complication types. For this research, we considered every case which has a hint of neurological complication and may be included with other disease types. A total of 22,782 cases were finally considered out of 25,284 where only patients who were alive during the rescue were considered. In Table 1, all the diseases that were recognized as neurological complications are listed. These diseases had the most frequent record in the rescue station's database. Moreover, we did a verification with experts to finalize this list.

**Table 1.** Diseases under neurological complication.

| Nr. | Diseases |
|-----|----------|
| 1 | Transient ischemic attack |
| 2 | Intracranial bleeding |
| 3 | Subarachnoid hemorrhage |
| 4 | Seizure |
| 5 | Febrile seizure |
| 6 | Status epilepticus |
| 7 | Apoplexy |
| 8 | Intercostal neuralgia |
| 9 | Retrograde amnesia (after trauma) |
| 10 | Retrograde amnesia (without trauma) |
| 11 | Neurogenic shock |

The vertical pre-processing was done to sort out columns that contains repetitive or irrelevant information not useful for our research. As an example, the Glasgow comma scale (GCS) score is calculated by observing the response of eye-opening, verbal communication and motoric functionality of a patient. In the dataset, details of every response were recorded with individual scores. We considered working only with the total GCS score which is quite able to indicate the level of consciousness of a patient. Further, removed some columns where data availability is minimum and does not make any significant impact. After all these pre-processing, a total of 340 attributes were remaining that can be used for feature selection.

*3.6. Feature Engineering*

Features are described as signals that encode information from raw data facilitating machine learning algorithms to classify an unknown object or estimate an unknown value [20]. Feature engineering is the process of selecting, manipulating, and converting raw data into meaningful features that can be used in ML algorithms. The rescue data we used were quite complex and had no attribute directly referring to neurological diseases. That is why we went through multiple filtering processes to finalize our features. A three-fold filtering method was used to select the features:

1.  At the first stage, we collected empirical knowledge from past research, and consulted with medical professionals regarding neurological complications. From this study, we have sorted out 105 features pertaining to neurological complications. Henceme of those features were not directly present in the dataset but rather created using one hot encoding

2.  Next step was to review the data trend of selected 105 features separately for neuralgic and non-neuralgic patients. In our observation, 39 features were found showing significant differences between patients and non-patients

3.  Recursive Feature Elimination with Cross Validation (RFECV) technique was applied at the last step to the features selected at the second step. RFECV identifies the most relevant and impactful features and removes those with minimal impact. Both random forest and gradient boosting algorithms were applied with RFECV to select the final features

Our filtering process concluded with only five features which are GCS score, Presence of trauma, brain injury, head discomfort and consciousness disorder. The other features including health vitals were not found to make any significant impact on the detection process of neuralgic complications. Because the values of the vitals for neuralgic patients

were quite similar to patients with other complications. In Figure 3, multiple bar plots of health vitals are drawn between patient and non-patient to prove our claim.

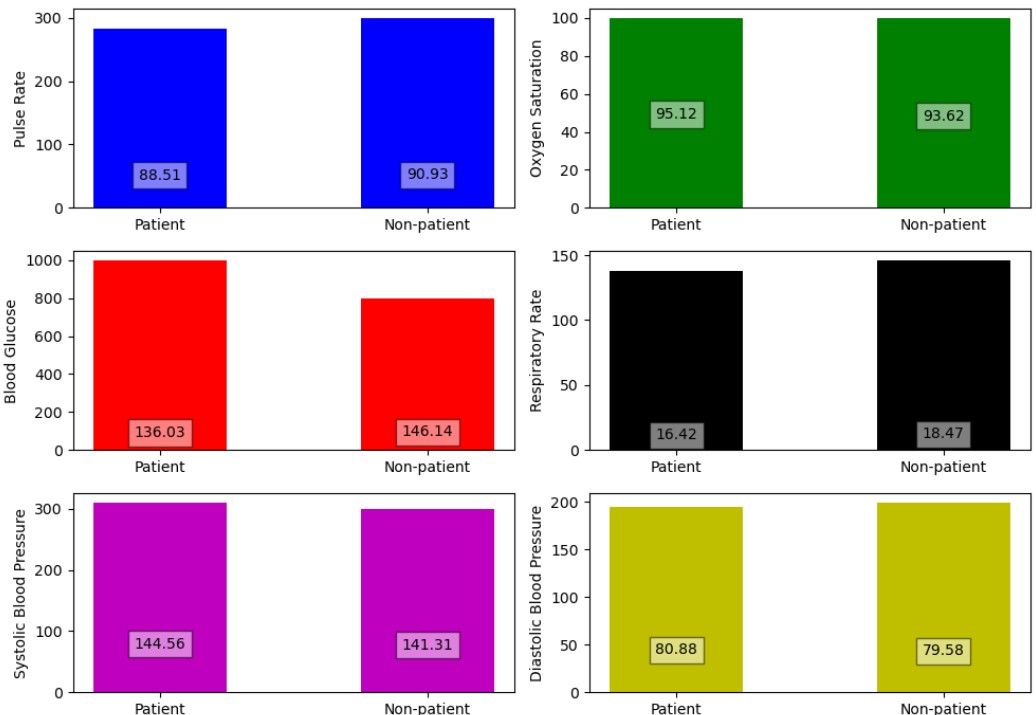

**Figure 3.** Comparison of Health Vitals between Neurological and Non-neurological patients.

### 3.7. Natural Language Processing

The use of only 5 features to identify a patient's neurological disorder is not practical, especially when the situation is sensitive, as can be observed in our case. As mentioned before, health vitals are not important indicators for neurological diseases. In hospital conditions, health professionals generally perform a variety of medical tests like CT scans, Electroencephalography, MRI ETC to identify neurological disorders, but in a rescue situation, the rescuers have no time to wait for performing these tests, or they don't even have the equipment support to do so. Here, we applied a novel natural language processing (NLP) technique to the text data of the rescue database to gather important attributes of rescue patients. The goal was to find out the common symptoms of a neurological patient in emergency situations and create new features.

Natural language processing (NLP) is known as a collection of computational techniques for automatic analysis and representation of human languages, motivated by theory. It is quite efficient when a given set of text information becomes increasingly difficult to disseminate by a human to discover the knowledge/wisdom in it, specifically within any given time limits [21]. Text parsing, an application of NLP, was used in our research to create some new features that can strengthen the performance of our ML model. At first, we merged all textual data relevant to neurological complications from the rescue database into a single file. Afterward, we created a second file containing keywords that is relevant to neurological complication. The keywords were also categorized into different groups to indicate their characteristics e.g., patient's current situation, neurological abnormality, the possibility of communication disturbance, presence of previous neurological illness, etc. To find out those keywords, we developed a word-counting algorithm that uses the text data file and checks how many times a word appeared in that file. We used Natural Language Toolkit (NLTK) library to design this algorithm. NLTK is one of the most popular platforms in the Python environment that can process human language data for application in statistical natural language processing. It contains text processing libraries for tokenization, parsing, classification, stemming, tagging and semantic reasoning [22]. Choosing the

right keywords was a complicated process, as some of the keywords are also common in non-neuralgic diseases that can decrease the detection accuracy. Hence, we developed a Python-based validation algorithm that checks whether a keyword is more relevant to neuralgic diseases rather than other diseases. The output of the algorithm was a validation score, which is simply a ratio of the percentage of time a keyword appeared in text data from neuralgic diseases to other diseases. Based on our observation of the data, we decided on a threshold score of 6 or above for a keyword to be selected. After parsing the selected keywords into the text file, we got three new features. In Table 2, all the features with their description are shown.

**Table 2.** List of finally selected features.

| Features | Value Range | Feature Description |
| --- | --- | --- |
| GCS | 3–15 | Type: Categorical<br><br>Score of Glasgo comma scale. It describes the level of consciousness in a person.<br><br>3 means almost no consciousness<br>15 means full consciousness |
| Neuro Abnormality | Binary | Type: Categorical<br><br>Indicates if the patient was having any sign of abnormality pertained to neurological conditions like: disorganized, body shaking, speaking without sense, or proper consciousness.<br><br>1 indicates presence of such characteristics<br>0 indicates the opposite |
| Communication Disturbance | Binary | Type: Categorical<br><br>Clear sign of neurological disturbance like: severe communication disorder, no feelings of anything, visual disturbance, no consciousness or concentration.<br><br>1 indicates presence of such characteristics<br>0 indicates the opposite |
| PreIllness | Binary | Type: Categorical<br><br>Any history of neurological pre-illness e.g., epilepsy, parkinson, meningitis etc.<br><br>1 indicates presence of such characteristics<br>0 indicates the opposite |
| Consciousness Disorder | Binary | Type: Categorical<br><br>Complaint from patient's side for consciousness disorder before the rescuer's arrived.<br><br>1 indicates presence of such characteristics<br>0 indicates the opposite |

**Table 2.** *Cont.*

| Features | Value Range | Feature Description |
|---|---|---|
| Pain or Discomfort at Head | Binary | Type: Categorical<br><br>Complaint from patient's side for any discomfort or pain in head.<br><br>1 indicates presence of such characteristics<br>0 indicates the opposite |
| Brain Injury | Binary | Type: Categorical<br><br>Presence of traumatic brain injury<br><br>1 indicates presence of such characteristics<br>0 indicates the opposite |
| Trauma | Binary | Type: Categorical<br><br>If the patient recently experienced any trauma<br><br>1 indicates presence of such characteristics<br>0 indicates the opposite |

### 3.8. Use of AI Model

Machine learning is a subset of Artificial Intelligence (AI) that has the capability to imitate human intelligence to analyze a set of data provided by users, acquire important insights, and then make informed decisions. The use of machine learning proved to be quite effective for datasets where the human brain is not capable to draw necessary insights due to the size of the data. There are a number of algorithms that can be occupied with our rescue dataset, but the goal is to find out the best algorithm for detecting health complications. Based on our findings on state-of-the-art research, we selected seven algorithms as below:

- Extreme gradient boosting (XGB)
- Support vector machine (SVM)
- Random forest (RF)
- Naive Bayes (NB)
- Logistic regression (LR)
- Artificial neural network (ANN)
- K-nearest neighbor (KNN)

### 3.9. Hardware Acceleration

To implement an AI model in a real-world scenario, a stand-alone hardware prototype capable to do fast inference is quite necessary. In our research project 'KIRETT', we are currently developing an FPGA-based wearable that can be used independently by a rescuer to get diagnosis support. The FPGA will be accelerated with an AI model that provides the best performance in rescue-based health complication detection. The wearable will provide the following benefits:

- Improved quality of care in rescue missions
- Increase diagnostic efficiency of rescuers through automation and contextual support
- Initialize quick treatments paths after complication detection

We selected PYNQ-Z2 FPGA as our stand-alone AI accelerator because of its energy efficiency, low latency, programming flexibility and high scalability with complex AI models.

This paper will provide a detailed analysis of our hardware prototype and deployment of the AI model in hardware at Section 5.

### 4. Development of Machine Learning Algorithms

The aforementioned machine learning algorithms were used in this step to develop a detection model that can indicate if a rescue patient is suffering from neurological disease or not. We used 'Pycharm' and 'Jupyter Notebook' as IDEs to work with Python-based development. For the data pre-processing and model optimization, 'Scikitlearn' library was used. The ML algorithms we chose consist of both core machine learning models and a deep learning model. Open-source software libraries like 'Keras' and 'Tensorflow' were used to design the deep-learning neural network model. Keras is implemented like an interface for the TensorFlow framework which allows developers to focus on the main concepts of deep learning, such as defining layers for neural network models without going through deep details of tensors, their shapes, and their mathematical details [23,24]. The final dataset we came up with after the data analysis part is the training dataset common for all machine learning models.

Support vector machine (SVM) is one of the most widely used supervised machine learning algorithms for data classification and regression modeling. This algorithm is mostly used for pattern recognition or classification based on either a priori knowledge or statistical information extracted from raw data [25]. The versatility of SVM algorithm in terms of choosing different kernel functions for decision function makes it quite advantageous for researchers in this domain, but based on our observation, it takes longer time for training than other algorithms when the dataset is large.

The classification principle of K-nearest neighbor is much simpler than SVM. It works by finding the distances between a query and all the examples in the data, selecting the specified examples that are closest to the query, and then voting for the most frequent label [26]. The flexibility of this algorithm for classification schemes is very useful when the detection parameter belongs to multi-modal classes, but sometimes the model gets expensive when unknown records need to be classified. Further, if the used features are noisy or ambiguous, the detection accuracy is very poor compared to other models [27].

Random Forest is a tree-based supervised machine learning algorithm that collects random samples from the training dataset and creates decision trees based on the observation from each sample. Then an internal cross-validation decides the best decision trees to produce prediction results [28]. The greatest advantage of using this algorithm is: it is capable of filling missing values in attributes with the most probable value while showing high performance due to the efficiency of the tree traversal algorithm, but there are examples where the model showed unstable behavior because the model is prone to sampling error and gives an optimal solution locally but not globally [27].

Extreme gradient boosting (XGB) is an efficient and scalable implementation of gradient boosting framework as mentioned in [29]. Currently, the model is quite popular among data scientists due to its features such as ease of use, ease of parallelization and impressive predictive accuracy. As stated in [30], the algorithm showed promising performances in biomedical sectors. The algorithm also has the ability to handle missing and unscaled data and provide efficient performance when the dataset is large. Based on our observation, the model takes a much shorter time for training and provides very good accuracy when the training dataset is free from outliers.

Logistic Regression (LR), an algorithm used for the classification problem, is also used in our research. This supervised algorithm is generally used to calculate the probability of occurring binary events based on prior observation of a data set. We also applied the Naive Bayes algorithm to our dataset because of its unique classification technique. It is based on the Bayes theorem with an assumption of independence between features [31]. In simple terms, it means that the classifier assumes that the presence of a particular feature in a class is unrelated to the presence of any other feature.

*Model Tuning*

Tuning a model means selecting the hyperparameters of the model in a way so that it maximizes the model's performance without overfitting, underfitting, or creating a high variance. There are many methods to choose the best hyperparameters of a model of which the most popular are—'Grid Search' and 'Random Search'. In the 'Grid Search' method, we created a grid of hyperparameters first for each machine learning algorithm based on the usual parameter values and then these tuning functions are applied to loop through the hyperparameters grid, exhaust all combinations of hyperparameter subsets and evaluate the model's performance using cross-validation. Finally, the subset of parameters with the best performance score is provided as an output. 'Random Search' has some identical characteristics to grid search, but it does not go through all predetermined subsets of hyperparameters. Instead, the method randomly selects a chosen number of hyperparameters from the provided grid and checks only those to find out the optimum sets.

To achieve the best possible performance for the neural network models, 4 different tuners: Grid Search, Random Search, Hyperband and Bayesian Optimization were used. Based on our empirical evidence and many different simulations of binary classification data, we chose a neural network with 2 hidden layers. Afterward, the 4 tuning methods went through a grid of hyperparameters e.g., the number of nodes in each layer, dropout rate, activation function, and optimizer learning rate, to find out their optimum values of them. Our target was to achieve the highest accuracy with a minimum loss rate.

## 5. Hardware Deployment

During a rescue mission, rescuers are always on a limited time schedule to detect health complications and initiate possible treatment paths. Hence, becoming time-efficient is one of the major requirements in a rescue situation. After developing the detection model with ML algorithms, our goal was to accelerate it in an FPGA-based hardware prototype that would provide not only the same accuracy achieved in previous steps but also be very time economical, i.e., small inference time. Programmable hardware like FPGA was chosen due to its flexibility in hardware-level compilation, reprogramming ability, and impressive application performance even while processing multiple workloads at the same time. The Summary of the acceleration steps can be written below:

1. Selection of software framework for FPGA
2. Compilation of the ANN model with an open-source machine learning compiler like Apache Tensor Virtual Machine (TVM)
3. Quantizing and accelerating the ANN model with Versatile Tensor Accelerator

### 5.1. Software Framework PYNQ

The criteria for choosing a software framework for FPGA is to reduce the design effort in hardware coding and to get easy access to reusable components in FPGA. PYNQ (Python Productivity for Zynq) is an open-source, Python-based framework that can be deployed on all Xilinx SoC devices. PYNQ features a software stack that resides on the PS portion of the Zynq RFSoC device and facilitates user interaction via a network connection and a standard web browser, e.g., chrome [32]. In addition, the PYNQ framework incorporates a monolithic pre-configured bitstream that contains built-in libraries and integrated Intellectual Properties (IPs) to support a range of hardware peripherals, easing the burden of non-FPGA experts and software programmers with little to no hardware expertise [33]. The key features of the PYNQ framework are listed below:

- Linux or Ubuntu-based operating system that provides an easy-to-use platform for developers to harness the advantage of power and flexibility of FPGA acceleration.
- Pre-installed Python libraries and integrated drivers to develop embedded systems faster using high-level Python-based programming.
- Development tools such as 'Jupyter Notebook' and Pre-built overlays for far more accessible features. The notebook provides a user-friendly interface to code and test

algorithms. The pre-built overlays provide access to the overlay functions for common tasks with the hardware accelerators.

Various FPGAs are configured with the PYNQ framework and a Yocto-based tool flow, like the low-cost Zynq architecture PYNQ-Z2 and PYNQ-Z1. Both are low-cost development boards designed explicitly for FPGA-based neural network deployment. After going through all the technical detail, we decided to use PYNQ-Z2 FPGA, shown in Figure 4, an attractive embedded platform for several reasons as below:

- Higher Processing and Performance: It features a dual-core ARM Cortex-A9 processor, which provides high performance for complex pre-processing on the input data [34] and manages workload more efficiently
- Flexibility and Customizability: The PYNQ-Z2 board provides a highly flexible and customizable platform for neural network deployment to optimize performance, improve accuracy, shorten inference time at lower power consumption [35], and customize neural network architectures for specific applications compared to using off-the-shelf hardware.
- Larger memory: The PYNQ-Z2 has 1 GB DDR3 RAM, making it more versatile and capable than the PYNQ-Z1 with 512 MB DDR3 RAM. The more significant onboard memory can be beneficial to store the data while working with large-scale data sets before feeding it into the neural network model. In addition, it provides a highly low-latency performance, a critical requirement for real-time NN applications [36].
- I/O options: The PYNQ-Z2 provides more I/O options to connect a wide range of external peripherals, such as medical equipment and user interface devices, for communication and easier integration with the FPGA with additional interfaces like SATA, PCIe and FMC connectors.

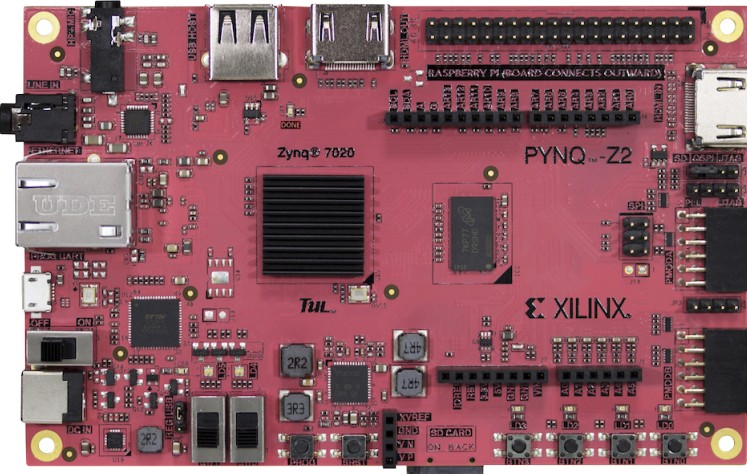

**Figure 4.** PYNQ-Z2 Development Board [37].

### 5.2. Machine Learning Compiler TVM

Apache Tensor Virtual Machine (TVM) is an open-source machine learning compiler framework [38], which optimizes the code to deploy machine learning models for various hardware platforms, including CPUs, GPUs, FPGAs, and specialized accelerators. Apache TVM accepts inputs from ANN frameworks such as TensorFlow or PyTorch. TVM support for model training is under development [39], so, currently, TVM does not support the training of models similar to most ANN compilers which have only limited support as the emphasis is on inference and therefore focus is shifted to effectively deploying models to target hardware in the best possible optimization. It is, therefore, imperative to train the model with a different platform before loading the pre-trained model on the TVM frontend.

Compiling the ANN model using TVM consists of multiple steps, as shown in Figure 5. The TVM compiler takes the high-level neural network model in the TensorFlow framework,

translates the model into a hardware description language (HDL), performs different stages of optimizations, and then generates a bitstream suitable for deployment into the FPGA.

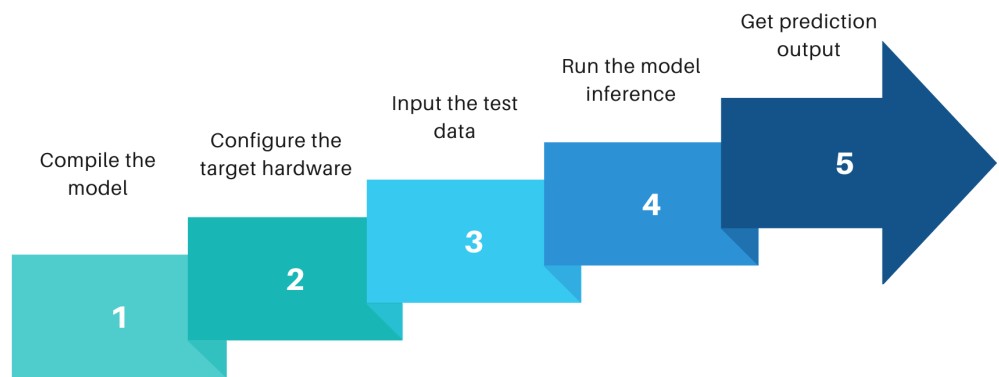

**Figure 5.** TVM Compilation Flow.

### 5.3. AI-Accelerator VTA

The Versatile Tensor Accelerator (VTA) is an open-source hardware accelerator designed by Apache TVM to enable the efficient execution of machine learning models on FPGA. In addition, VTA supports quantization, a key feature to adapting the models efficiently for deployment in specific hardware resources. For efficient deployment on FPGA using the VTA, its backend workflow for ANN deployment involves designing and training the model, performing quantization, compiling the model, configuring the hardware, taking input for testing, running inference, and the final post-processing stage for providing output. The advantage of VTA is that it is fully customizable, i.e., the hardware intrinsic properties, memories, and data types can be customized to adapt to the hardware backend requirements [14].

Quantization of the ANN model is a crucial step for successful deployment. It is the process of reducing the bit width of a deep learning model's weights and activation functions by sharing parameters, decreasing hardware resource usage, and consequently optimizing the model for the target FPGA [40]. Apache TVM can convert a high-level ANN model into a deployable quantized module on a range of hardware platforms. It enables the quantization and efficient execution of models from deep learning frameworks with almost the same accuracy or just a little decline from the original accuracy. The compiler uses an 8-bit integer quantization to efficiently increase the resources of specific target hardware platforms.

To optimize the neural network for specific hardware and thereby reduce the memory footprint, Apache TVM quantizes the 32-bit float values to 8-bit integers. In addition, it utilizes VTA to perform further quantization to optimize the ANN models by creating a balance between model accuracy and FPGA resource constraints. The purpose of such a quantization process is to enhance time efficiency and better resource utilization since quantization affects the performance of a model as a function of the model depth [40]. Other than quantization, VTA also performs other operations, including the fetch, load, compute, and store, that work together to manage the data flow and optimize the performance of the inference process.

The modules inside VTA, as shown in Figure 6, have specific work functions as below:

1. The fetch module loads a stream of instructions from the dynamic random-access memory (DRAM), decodes them and transfers them into the corresponding FIFO queues.
2. The load module loads the DRAM's input, weight, and bias tensors into data-dedicated on-chip memories.

3. The compute module performs various computations involving matrix multiplication and activation functions. It also loads data and micro-op kernels from the DRAM into the register file of on-chip memory and the micro-op (UoP) cache, respectively.
4. The store module then retrieves the results generated by the compute module and sends them back into the DRAM.

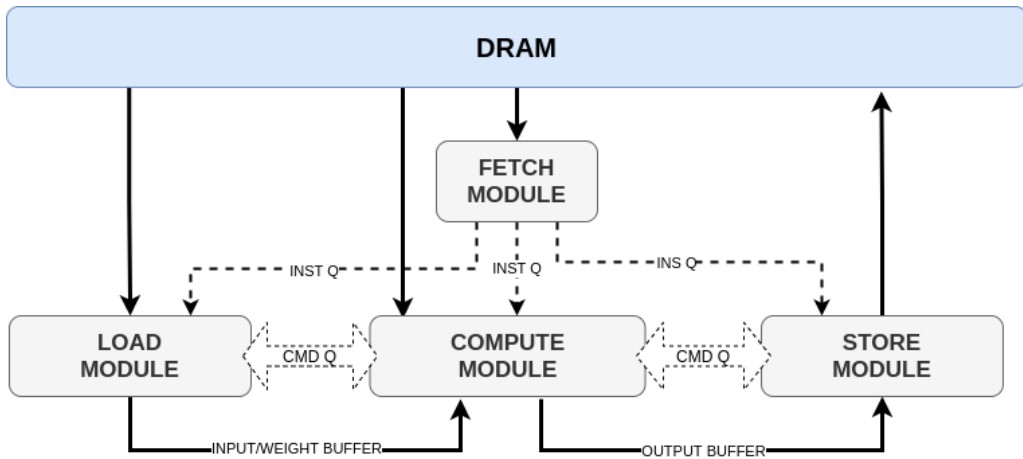

**Figure 6.** Versatile Tensor Accelerator (VTA).

## 6. Results and Discussions

The primary focus of our developed supervised machine learning models is to predict a category or class from the input feature vectors. Feature vectors are known as n-dimensional vectors of numerical features that describe the characteristics of a sample object. Each row in the training dataset we developed before represents a feature vector with a length of 8, which is the number of our finally selected features. Classification-based detection algorithms normally have a categorical output like a "disease" or "no disease", "1" or "0", "True" or "false". We used 1 or 0 for categorization and put this label in the last column of our dataset. If a rescue patient is diagnosed with neurological diseases, the label is set to 1, and it is 0 if the patient is healthy or suffering from other health complications rather than neurological ones. We have considered (randomly) 23,090 rescue cases of non-neurological patients in the dataset to make the total dataset length of 45,872 feature vectors. The whole dataset has been split into two parts where one part is the training dataset and the other one is the testing dataset with a ratio of 70:30, respectively. The training dataset was fitted to each machine learning algorithm to build and train the model and teach how the expected output should look like. The purpose of testing the dataset is to evaluate the performance of each model with unknown feature vectors. We also performed three-fold cross-validation to check the model's performance by using a portion of training data. Cross-validation is a data resampling method to assess the generalization ability of predictive models and to prevent overfitting [41]. In this validation process, we trained the machine learning models with the subset of training data and then evaluate their performance with a grid of hyperparameters. The cross-validation technique applied here is K-fold cross-validation, where the random sampling is done in a way so that no two test sets overlap with each other. Performance of the selected algorithms in each fold of cross-validation can be seen from Table 3.

**Table 3.** Cross-Validation Performance of ML models. XGB showed the best performance in terms of cross-validation accuracy followed by the neural network model. The random forest model had the lowest standard deviation of 0.11% in accuracy across cross-validation folds. All models except Naive Bayes showed a mean accuracy of more than 80%.

| Classification | Fold 1 | Fold 2 | Fold 3 | Mean CV Score | STD Accuracy |
|---|---|---|---|---|---|
| XGB | 86.58 | 85.84 | 85.20 | 85.87 | 0.56 |
| KNN | 84.85 | 84.03 | 83.26 | 84.05 | 0.65 |
| Random Forest | 85.13 | 85.16 | 84.92 | 85.07 | 0.11 |
| SVC | 86.08 | 85.17 | 84.69 | 85.31 | 0.58 |
| Logistic Regression | 86.25 | 85.47 | 85.01 | 85.58 | 0.51 |
| Naive Bayes | 81.09 | 81.57 | 73.07 | 78.58 | 3.90 |
| ANN | 86.43 | 85.58 | 85.14 | 85.72 | 0.54 |

　　　To visualize the stability of the models' performance through cross-validation folds, a box and whisker plot can be used. Such a plot is also used to show how the data are distributed and if there are presence of outliers in the dataset. In Figure 7, the spread of the accuracy scores across each cross-validation fold for each algorithm is shown using a box and whisker plot. From the figure, it can be seen that the Naive Bayes model has a bigger box size, as can be also understood from its high standard deviation of 3.9% between folds' accuracy, but looking at the box shapes of other models, we can easily infer that the accuracy achieved through cross-validation folds is balanced and carries no outliers.

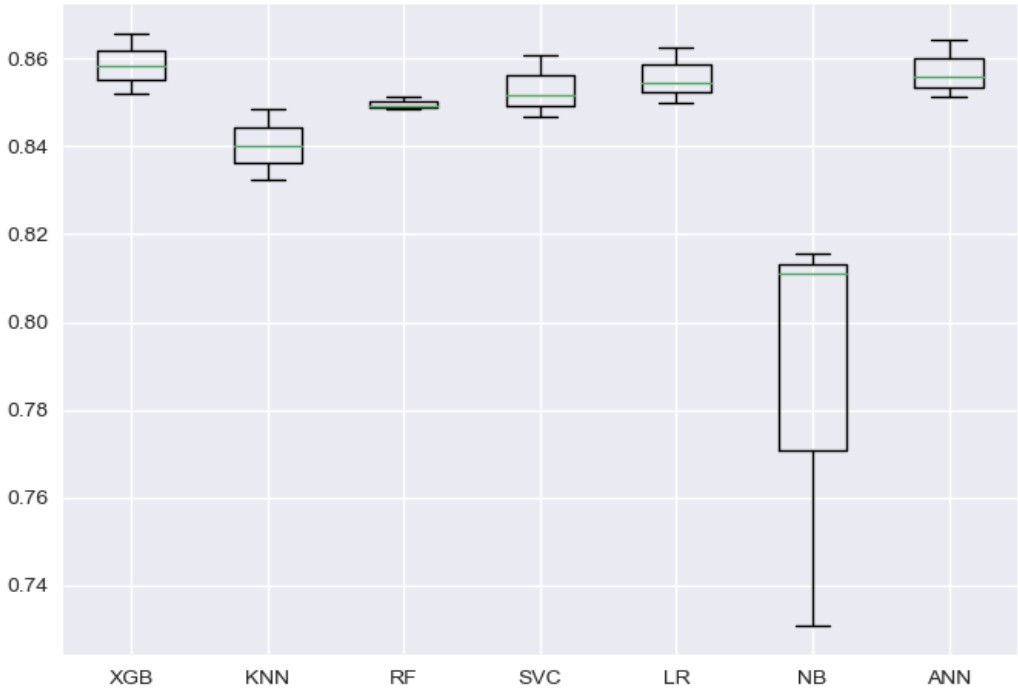

**Figure 7.** Cross Validation Accuracy of Each Fold for All ML Algorithms.

　　　A three-fold cross-validation technique was applied to the training dataset with 'GridSearch' and 'RandomSearch' methods to finalize the optimum hyperparameters of each ML model. After training the models with the selected hyperparameters, we started the evaluation process with the test dataset by plotting confusion matrices for each model.

A confusion matrix represents the prediction summary in matrix form. It consists of four different combinations of predicted and actual values, as can be seen in Figure 8. Statistical measures like accuracy, sensitivity, specificity, precision, and F1 score were derived from the matrix as performance metrics to derive the model's effectiveness and performance.

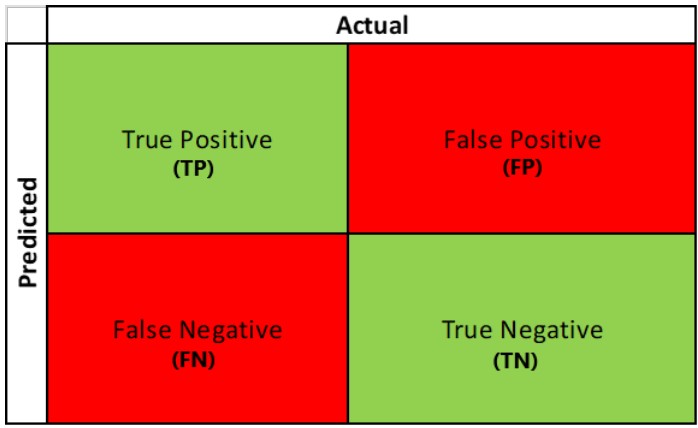

**Figure 8.** Confusion Matrix.

A short definition of the performance metrics with their calculating formula is given below:

- Sensitivity: The ability of a machine learning model to predict true positives.

$$\text{Sensitivity} = \frac{TP}{TP + FN} \tag{1}$$

- Specificity: It describes how well a classifier can detect the true negatives.

$$\text{Specificity} = \frac{TN}{TN + FP} \tag{2}$$

- Accuracy: Accuracy can describe a model's capability to do accurate prediction overall predictions.

$$\text{Accuracy} = \frac{TP + TN}{TP + TN + FP + FN} \tag{3}$$

- Precision: It is the ability of a classifier to identify what proportion of positive results are actually positive.

$$\text{Precision} = \frac{TP}{TP + FP} \tag{4}$$

- F1-Score: F1-Score measures a model's accuracy on a dataset.

$$\text{F1-Score} = \frac{2TP}{2TP + FP + FN} \tag{5}$$

With these matrices, we made a performance comparison of all the models to find out the best. Evaluation of all the detection models on neurological complications showed us accuracy in the range of 80–86%. Among the classification techniques, the best accuracy was shown by the ANN algorithm, with 86.62%. The model has a sensitivity of 88.17% and a specificity of 85.06%. The next best accuracy of 85.85% was achieved by Extreme gradient boosting. Random forest algorithm, Logistic regression and Support vector machine showed a similar performance with an accuracy of around 85%. All models showed an accuracy of over 80% while Naive Bayes had the lowest of 80.65%. The detailed performance of all the algorithms is shown in Figure 9.

| Classification | Accuracy | Sensitivity | Specificity | F_1 score | Precision |
|---|---|---|---|---|---|
| ANN | 86.62 | 88.17 | 85.06 | 86.82 | 85.51 |
| XGB | 85.85 | 88.21 | 83.49 | 86.14 | 84.16 |
| Logistic Regression | 85.49 | 86.60 | 84.41 | 85.51 | 84.45 |
| SVM | 85.34 | 86.42 | 84.29 | 85.37 | 84.35 |
| Random Forest | 85.10 | 81.58 | 88.58 | 84.47 | 87.58 |
| KNN | 84.52 | 79.08 | 89.96 | 83.63 | 88.73 |
| Naive Bayes | 80.65 | 70.16 | 91.14 | 78.38 | 88.78 |

**Figure 9.** Performance of Detection Models.

The XGBoost model yielded the best sensitivity of 88.21% followed by ANN with 88.17%. NB model showed a specificity of 91.14% and precision of 88.78% which is better than other models while the sensitivity was quite poor—70.16%. ANN has again shown the best performance in terms of F1 score—86.82%. The performance of the XGB model was quite comparable to ANN. It achieved an accuracy of 85.85% with sensitivity and specificity of 88.21% and 83.49% respectively. A visual representation of the performance for all the models is depicted in Figure 10.

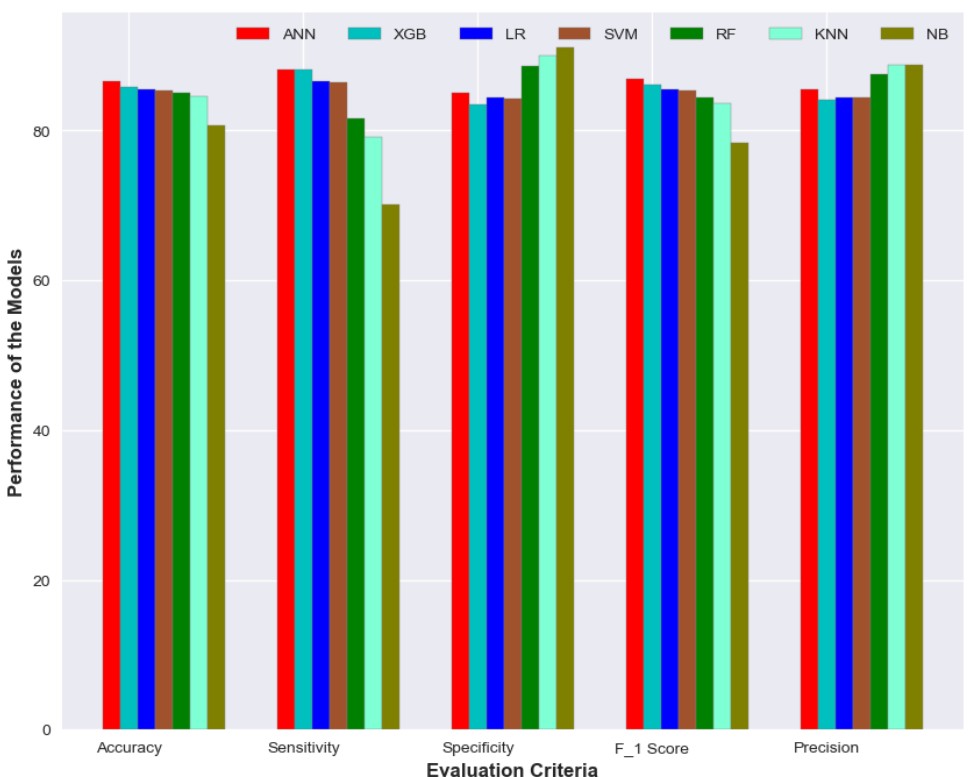

**Figure 10.** Performance Comparison of the AI Models on the Selected Evaluation Criteria.

To find the best-performing algorithms, we cannot just rely on the evaluation matrices. Because for any imbalanced data set, the model will always predict the majority class which is unexpected. A receiver operating characteristic (ROC) curve can be used to assess if a dataset has such critical points. ROC curve is a graph representation of a classification model at various classification thresholds, while Area-under-Curve (AUC) tells us the

degree of measure of separability between binary classes. The two parameters used to draw the curve are: the false positive rate and the true positive rate of predicted classes. Generally, ROC-AUC curves can judge the discrimination ability of various statistical methods that combine various clues, test results, etc. for predictive purposes [42]. The value of AUC indicates the classification performance of a model. When it is 1, it means the classifier can correctly distinguish between all the patients and the non-patients. A value of 0 indicates that the classifier would predict all the non-patients as patients and the patients as non-patients. Below in Figure 11, an ROC-AUC curve is drawn to summarize the performance of all the models depicting the ability of each model in the detection of positive cases.

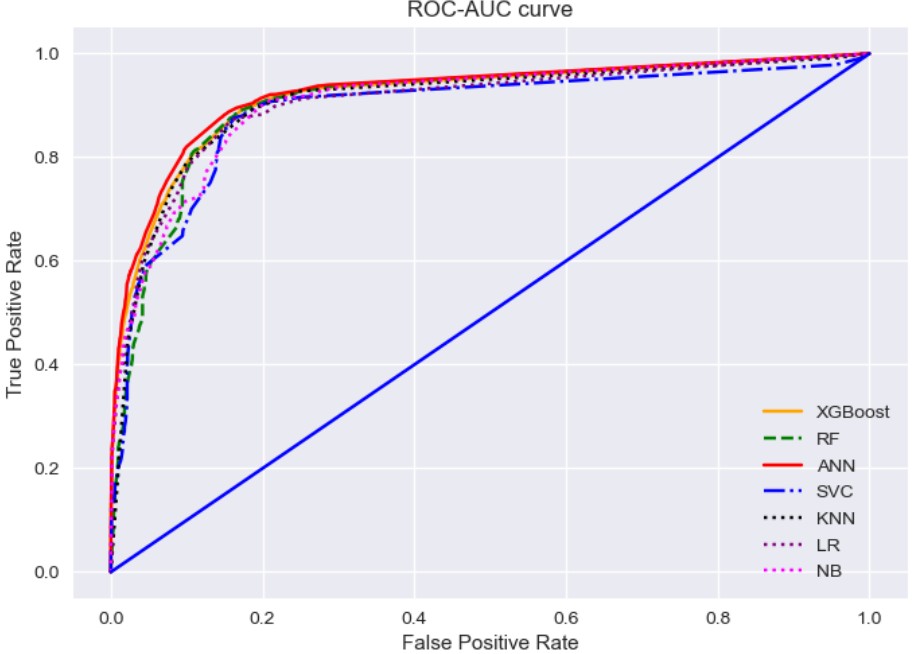

**Figure 11.** ROC-AUC curve to compare ML models' performance.

The ROC-AUC curves for all the models are almost entangled with each other as their performances are quite similar. The red solid line in the curve is representing the AUC score of the ANN model, which has a slight edge over other models. It is because the model has the highest AUC score of 0.924 which indicates that the ANN model has the best ability to detect true positive cases. The AUC score of the support vector model is the lowest—0.890 as depicted with the blue dash-dot line, but the value is quite close to ANN's score and surely can demonstrate comparative performance in the detection of true positive cases.

An important observation we made from all the classification models is: not all the features among the selected 8 are significant. Their importance and ranking are varied from model to model. We used the 'Feature Importance Score' to identify how each feature makes an impact on the detection/classification accuracy. A Scikit-learn library 'Permutation_Importance' was implemented in our algorithm to assign a score for all the input features for a given model, where the scores simply represent the "importance" of each feature. A higher score indicates that the specific feature will have a larger impact on the model for classifying patients and non-patients. In Figure 12, feature importance scores of all the models are depicted for better understanding.

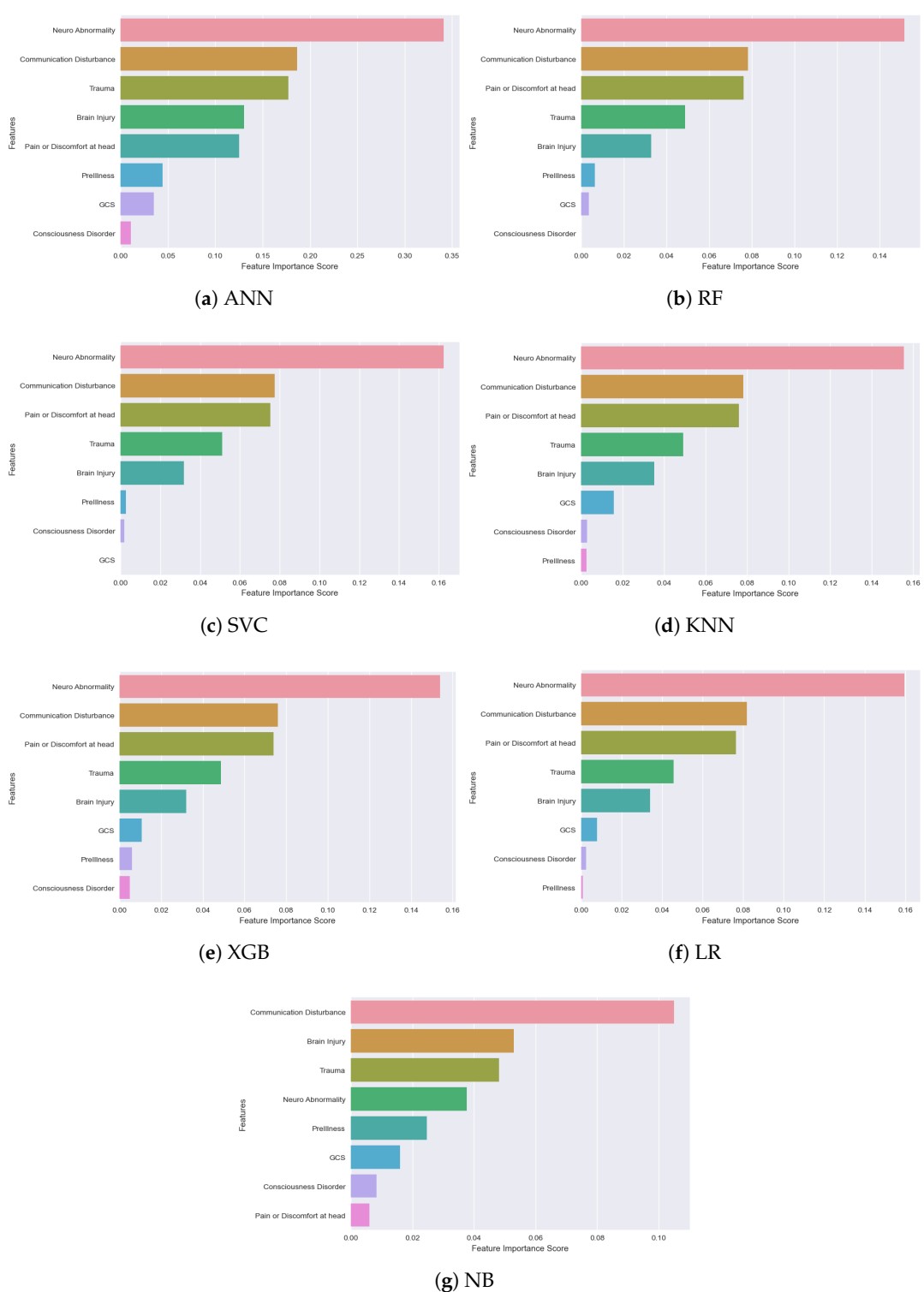

**Figure 12.** Impact of the selected features in all ML models.

The observation from the above figure states that almost every model has their own feature importance ranking. Based on our best-performing model ANN, the three most important features are—'Neuro Abnormality', 'Communication Disturbance' and 'Trauma', but for the XGB model, 'Pain or Discomfort at head' is the third most important feature instead of 'Trauma'. 'Communication disturbance' and 'Neuro Abnormality'—seem to be the most important two features in all the models except the Naive Bayes model. 'Communication Disturbance', 'Brain Injury' and 'Trauma' are considered the most important

features of this model. The results pointed out how important it is to implement NLP to extract new features from the text data of the rescue database. All this is generally recorded by the rescuers at emergency sites from their personal observation. Our models suggest that the first observation rescuers need to perform for detecting neurological complications is to find if the patient has any impairments in vision, speech, or movement. Other signs to be observed are: the patient's choice of words, capability to control the body and overall responsiveness. As for neurological abnormalities, rescuers need to check signs of seizure attack, the shape of the mouth, tongue bite, uncontrolled urination, etc. in patients. It is also important to take note of historical records of patients like if the patient had any trauma or brain injury recently, or whether he or she has any past record of neurological complications like epilepsy or brain tumor. Such observations should be given utter importance and passed to the machine learning-based system as data immediately for quick diagnosis. In our findings, GCS score did not make a high impact in any of the models. One reason could be that, due to the nature of rescue situations, it is quite difficult to perform such a test or record the score correctly.

Another goal of our research was to calculate the inference time for each model with test data to understand if the model will be time efficient in real-world scenarios. We did the calculation by passing a single test data to each of the models and then record the time of detection. The result can be seen in Table 4.

**Table 4.** Inference time for ML models.

| Model | Inference Time (Seconds) |
|---|---|
| XGB | 0.002992 |
| ANN | 0.048838 |
| RF | 0.006981 |
| SVC | 0.001994 |
| KNN | 0.001987 |
| LR | 0.001006 |
| NB | 0.000960 |
| TVM-VTA | 0.013271 |

The table showed that the FPGA accelerated ANN model with Apache TVM-VTA had an inference time of only 13 ms and was actually faster than the deep learning model compiled with a computer CPU. The outcome is quite significant because it ensures that the 8-bit quantization process handled by TVM-VTA resulted in a very time-efficient detection model. It can be an important breakthrough in AI-based hardware acceleration as the FPGA prototype was not only just time economic but also showed no distortion in terms of accuracy compared to the original ANN model.

*Limitations and Challenges*

The biggest challenge we faced during our research was sorting and filtering the data of rescue cases. The rescue database was full of null values and sometimes meaningless data. This is quite understandable as it is very difficult to collect data in emergency cases. Based on our findings, in more than 80% of the cases, the patients or related people cannot provide enough information. Sometimes, the patients do not even want to disclose health information or are uncomfortable answering health-related questions to rescuers. The health vitals taken are often manually recorded, as the sensors don't work well when a patient is unstable. During our data analysis, we considered those data limitations and used extensive data mining and pre-processing steps as explained in Section 3.

We also predicted potential challenges in the hardware part. The wearable we wanted to develop by deploying an AI model in FPGA-based hardware may face some limitations in the future as below:

- Battery life and reliability can be considered as the main challenge for stand-alone FPGA-based hardware. Further research is needed to find a reliable solution, e.g., exploring Bluetooth technology instead of WiFi for patient data collection, use of a buck-boost converter on expanding wearable battery life, etc. Further, in rescue contexts, the reliability of a healthcare device is quite important. In a massive accidental situation, where the rescue environment gets quite harsh, it is quite challenging for a hardware device to function properly unless it has high robustness.
- Interfaces to medical devices manufactured by different companies for collecting data are often a great challenge. Generally, the manufacturers have different protocols for data transmission and sharing as no universal standard exists. Moreover, the transmitted data format is not unique and depends on the compatibility of the device. Hence, in the future, the wearable may face challenges to integration with different medical devices and generation of test data for the AI model.

## 7. Conclusions

Artificial intelligence has the potential to fundamentally alter practices of the health sector in the near future. In the area of neurological diseases, where health vitals are not the most contributing features for prognosis, AI has drawn immense attention from researchers to be used as a supporting tool for better diagnostic and prognostic evaluation. The early and precise diagnosis of neurological disorders is an incredibly challenging task, especially in rescue situations, for both healthcare professionals and data scientists. In this paper, we have provided a comprehensive overview of these challenges and a novel NLP-based solution to develop a concrete detection model. The features and relevant keywords we extracted from the dataset could come out quite resourceful for rescuers in their future missions. As the scope of machine learning and artificial intelligence-based diagnostic systems is extending rapidly every year, the possibility of smarter and faster healthcare tools is obvious. Our proposed neural network-accelerated FPGA is already a good example of it due to its time-economic and efficient performance. In the future, other scalable and programmable devices like ASIC can be used to develop a smart wearable for rescuers. The authors of this paper, as a part of the research project 'KIRETT', are currently working on expanding the research scope by integrating other health complications e.g., psychiatric or cardiovascular diseases in the detection process.

**Author Contributions:** Conceptualization, A.S.A., methodology, A.S.A., Data analysis A.S.A., Model development A.S.A., Model tuning A.S.A., Validation A.S.A., Model evaluation A.S.A., Hardware selection A.M.E., Hardware deployment A.M.E. and A.S.A., Prototype efficiency A.M.E. and A.S.A., Investigation A.S.A. and A.M.E., Resources A.S.A. and R.O., Writing—original draft A.S.A. and A.M.E., Writing—review & editing A.S.A., R.O. and A.M.E., Supervision R.O., Project administration R.O. All authors have read and agreed to the published version of the manuscript.

**Funding:** The ongoing research was financially supported by the Federal Ministry of Education and Research, Germany. The research has been supported by KIRETT project coordinator CRS Medical GmbH (Aßlar, Germany), and partner organization mbeder GmbH (Siegen, Germany). The authors would like to thank the associative partners of the project: Kreis Siegen–Wittgenstein, City of Siegen, the German Red Cross Siegen (DRK) and the Jung-Stilling-Hospital in Siegen.

**Informed Consent Statement:** Patient consent was waived because this study used the data from another resource and informed consent was obtained from all subjects involved in that study.

**Data Availability Statement:** The data were collected from Rescue Station, Siegen–Wittgenstein. The data are subjected to privacy issue and can only be accessed with prior approval.

**Conflicts of Interest:** The authors declare no conflict of interest.

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
