# Peer review of "Time-Efficient Identification Procedure for Neurological Complications of Rescue Patients in an Emergency Scenario Using Hardware-Accelerated Artificial Intelligence Models"

_algorithms, doi:10.3390/a16050258_

Round 1
Reviewer 1 Report
The authors present a step-by-step analysis of developing multiple machine learning models that can facilitate the fast identification of neurological complications in general.
Positive aspects:
- + The paper addresses a relevant problem.
- + The description of the work is generally clear. The steps and parts are comprehensible and in most cases clearly motivated. They follow a scheme of general/good practice (that’s how people do it).
Negative aspects/open issues:
- - Language comprehensible, but sometimes a little sloppy; some grammatical errors (e.g., with articles) and typos (e.g., check the name of your department).
- - I believe that the use of decentralized and/or stand-alone hardware could be beneficial in real rescue scenarios. However, the use case did not really convince me. In practice, the inference time of the models is probably not a critical factor in compared to the time needed for data collection. The hardware part needs to be better motivated to avoid the impression that you only did it because you can.
- - If you used cross-validation for the final performance assessment (Figure 8) please specify the number of folds and report the statistics (mean and standard deviation) in Figure 8.
Please address these points when revising the paper.
Language comprehensible, but sometimes a little sloppy; some grammatical errors (e.g., with articles) and typos (e.g., check the name of your department).
Author Response
Dear Reviewer,
Thank you very much for your review. It certainly helped me to improve my paper. My corrections and response can be seen in the attachment.
Best Regards,
Abu Shad Ahammed

Reviewer 2 Report
The article "Time-Efficient Identification Procedure for Neurological Complications of Rescue Patients in Emergency Scenario Using Hardware Accelerated Artificial Intelligence Models" presents an interesting approach to identifying neurological complications in rescue patients using hardware-accelerated artificial intelligence models.
However, there are several areas where the article could be improved:
The article lacks a clear and concise introduction that outlines the problem and the significance of the proposed solution. The authors should provide a more detailed background on the current state of neurological complications identification in rescue patients. They should also explain how their proposed approach improves existing methods.
Furthermore, the methodology section should provide more detailed information regarding the hardware and software used. The authors should provide more information on the specific hardware and software used to implement their artificial intelligence models. Also, authors should provide information on the datasets they used for training and validation.
The results section could be improved by presenting more detailed information on the proposed approach's performance. The authors should provide more information about the accuracy, precision, and recall of their models. In addition, authors should provide any limitations or potential sources of error in their approach.
I believe the article could benefit from a more detailed discussion of the implications and potential applications of the proposed approach. The authors should provide more information on how their approach could be used in real-world rescue scenarios, as well as any potential limitations or challenges that may arise.
In summary, while the article presents an innovative approach to identifying neurological complications in rescue patients, it could be improved in several areas.
Author Response
Dear Reviewer,
It's been a pleasure to receive your review. The review points have really helped to modify my paper in a better way. In the attachment below, you will find my response with corrections.
Best Regards,
Abu Shad Ahammed
